# The Dilemma of Conscience: From Paul and Augustine to Mencius

**Wei Hua**

Yuelu Academy, Hunan University, Changsha 410082, China; weihua@hnu.edu.cn

**Abstract:** Krister Stendahl's article, "The Apostle Paul and the Introspective Conscience of the West", argues that Paul has a "robust conscience" both before and after his conversion. Martin Luther misinterprets this as a "plagued conscience" in accordance with his own religious experience, and this misinterpretation can be traced back even to Augustine. This paper examines the context for the ancient Greek and Hellenistic theory of conscience, in order to understand Augustine's transformation of Paul's doctrine of justification by faith and the consequent discovery of the concept of introspective conscience in Western intellectual history. This paper also clarifies aspects of Augustine's "plagued conscience", which it analyses across two stages: the first after the descent of grace but before the conversion of a believer, and the second after conversion. In the first stage, Augustine implies a continuous spiritual conflict between good will and evil will within the inner self; however, in the second stage, the inner self experiences a deeper spiritual struggle, owing to its certainty of God's predestined plan alongside its uncertainty over personal salvation. The concept of introspective conscience has shaped the deep consciousness of sin for many Western Christians. This paper compares Pauline and Augustinian conscience with the same concept in the Confucian author Mencius. For Mencius, conscience is self-sufficient even in the earliest stages of its development and does not require the support of God's grace or the power of Heaven. The constant expansion of Mencius's operative conscience is sufficient for self-cultivation and the correction of the distorted world.

**Keywords:** Paul; Augustine; Mencius; conscience; *Romans*; Luther





## 1. Introduction

In both Chinese and Western moral philosophy, the concept of conscience has a long tradition. At least as far back as Homer, Western texts contain vocabulary that refers to conscience. Subsequently, this vocabulary was maintained and even extended in Greco-Roman drama and philosophy and entered canonical Western Christian texts through Paul's letters. However, Augustine's interpretation of conscience placed a profound emphasis on the consciousness of guilt in the Western Christian tradition. In Confucian texts, the word for conscience, *liangzhi* (良知), first appeared in writings attributed to Mencius. After a wide range of elaboration in Neo-Confucianism, Wang Yangming adopted the concept as the essence of his "doctrine of the extension of conscience" (*zhi liangzhi*, 致良知). Other Confucian philosophers also considered conscience as a significant concept; Mou Zongsan (1909–1995), a representative of Hong Kong/Taiwan Neo-Confucianism, put forward a theory of "negation of conscience" (*liangzhi kanxian*, 良知坎陷). The synonymity of these Western and Confucian concepts is evident in the common translation of the Greek *syneidesis*, Latin *conscientia*, and English conscience into the Chinese *liangzhi* or its synonym *liangxin* (良心). From a theoretical perspective, the concepts of conscience in the works of Paul, Augustine, and Schleiermacher have been studied comparatively with writings on conscience by Mencius, Zhu Xi, and Wang Yangming (Zhang 2015; Xie 2018; Kern 2012a, 2012b).

In 1963, Krister Stendahl (1921–2008) published his well-known article, "The Apostle Paul and the Introspective Conscience of the West". Stendahl was a Lutheran Swedish theologian who specialized in the New Testament. After receiving his doctorate from Uppsala

University, he later became dean, professor, and professor emeritus at Harvard Divinity School. In 1984–1988, he was even appointed as the Church of Sweden Bishop of Stockholm. In 1976, Stendahl published his book *Paul Among Jews and Gentiles and Other Essays* with the above-mentioned article as its core chapter. In it, he argues that Martin Luther, and even Augustine, misinterpreted Paul's evaluation of his own conscience. For Stendahl, the introduction of the doctrine of justification by faith supports Paul's contention that Gentiles were not required to abide by the Jewish Law; instead, they could only enter into the covenant that God had originally established with the Jews by their faith in Jesus Christ. In Stendahl's interpretation, Paul never experienced any moment of inner disturbance from sin, but abided by the Jewish Law and always had a "robust conscience"; however, Luther, affected by his personal experience of faith, misinterpreted Paul's conscience as a "plagued conscience". This enabled Luther's use of Paul's doctrine of justification to support his own position that human beings might only begin to attain peace for themselves after being justified by faith. Stendahl then traces this misinterpretation back to Augustine, often regarded as the discoverer of this hermeneutic tradition in the West and whose *Confessions* some consider to be the first major text in the history of the introspective conscience. Stendahl attempts to argue that Augustine's new interpretation of conscience sprang from a context far removed from that of Paul and the missionary motive behind his letters. This, Stendahl contends, resulted in consistent misunderstandings of these letters by Augustine, at the same time shaping the emergent concept of consciousness of guilt in Western philosophy. Since Augustine was also regarded as the discoverer of the "inner self" (Taylor 1992, pp. 127–42; Cary 2000), the introspective conscience or the "plagued conscience" became a precise and typical characteristic of the inner self.

In Confucian philosophy, Mencius is usually regarded as the originator of conscience as a concept. This is related simply to the use of *liangzhi* for the first time in his writings, where it denotes a kind of moral knowledge that human beings are born with. This innate moral knowledge—which Mencius's texts describe as a kind of "knowing without thinking" (*bu lü er zhi*, 不虑而知)—supported a doctrine of good human nature, which profoundly shaped the basic form of Confucian philosophy. Mencius, like Paul, was confident that this universally granted conscience could be constantly expanded. The Confucian doctrine of inner sagehood (*neisheng*, 内圣), which Mencius had a role in establishing, held that individuals could realize their moral cultivation inwardly. This moral cultivation could then be projected outwardly as "the protection of all within the four seas" (*baosihai*, 保四海), contributing toward peace and prosperity for one's family, country and, ultimately, the whole world.

Comparing the views of Paul and Augustine on conscience with that of Mencius clarifies key but subtle differences between the Western Christian and Chinese Confucian traditions in thought on human nature and moral psychology. In this article, I argue firstly that Paul inherited a concept of conscience shaped by the literature of the ancient Greek and Hellenistic period. This conscience is a kind of moral cognition capable of judging whether the behavior is good or evil. Paul's doctrine of justification by faith originally served to enable Gentiles to join in the covenant between God and the Jews. The missionary context in which Paul was writing had dramatically changed by the fourth and fifth centuries, so that Paul's letters, which had concentrated on collective salvation among the Jews and the Gentiles, were repurposed to argue for the salvation of particular individuals. Secondly, I argue that Augustine's introspective conscience is closely associated with the soul's self-judgment and self-condemnation of its own moral defect, giving rise to the interpretation of a "plagued conscience". This association originates from the incapability of the soul, in Augustine's thought, to will and do good without God's grace, due to the corruption and division of human will. Augustine's conscience is, therefore, more than a moral evaluation of good and evil behaviors in the ancient Greek sense. I further argue that Augustine discerned two stages of introspective conscience: before conversion and after conversion. The stage before conversion manifests as an internal conflict of will, described in the conversion narrative of *Confessions*, whereas the stage after conversion manifests as an uncertainty of

one's salvation within the framework of God's foreknowledge and predestination. Finally, I argue that, for Mencius, conscience that is directly endowed by heaven is always "robust". With its continuous extension, conscience enables human beings to undertake internal self-cultivation and motivates their external moral actions. In this sense, Mencius's evaluation of his own conscience is much closer to Paul's, and his "extensive conscience" is never a plagued one, but approaches the role of an "operative grace" that appears in Augustine's later works.

## 2. Paul on Conscience: Robust or Plagued?

Etymologically, conscience is derived from the Latin conscientia, which commonly translates to the Greek *syneidesis*. The Greek noun *syneidesis* is derived from the verb *syn-oida*, a compound of *syn-* and *oida* (*eidenai*), which means knowing something immediately or intuitively, as opposed to acquiring knowledge through reasoning (*noein*). This immediate or intuitive knowing underpins the basic meaning of three derivative nouns: *syneidesis*, *synesis*, and *syneidos*. The Latin *conscientia*, a direct translation of *syneidesis*, is a compound of *con-* and *scientia*, which is derived from the verb *scire*.

Conscience receives discussion in ancient Greek literature. As Jed W. Atkins has argued, lines 395–398 of Euripedes' *Orestes* deal with conscience, where *synesis* has three discernible features: (1) its verb *synoida* means to know or to recognize; (2) it has a reflexive function toward the subject; and (3) it refers to a kind of violation of moral laws. This sits within the bounds of four main ancient interpretations of conscience, including the soul's intellectual capacity, a subject who makes moral judgments, a faculty of the soul, and a cognitive action (Atkins 2014, pp. 1–22).

Changes in the meaning of conscience from the Classical to the Hellenistic period have been explored by Don Marietta. He argues that ancient Greek writers such as Euripides, Xenophon, Plato, and Aristotle, among others, use *synoida* mainly to denote consciousness or awareness of something and that its objects can be moral or non-moral. In the writings of Democritus and Chrysippus, *syneidesis* was not used in an ethical sense, and after those authors, the term disappeared until the beginning of the Christian era. However, the two terms *syneidos* and *syneidesis* shared the same meaning, and it is noteworthy that some authors preferred the use of one over the other. The flourishing of Hellenistic ethics fostered an internalized and individualized development of the concept of conscience. Although the Greek-speaking Stoics rarely used the term *syneidesis*, the Latinate Stoics' use of *conscientia* reflected a conception of *syneidesis* that could apply to moral and non-moral objects; there was a common understanding in the Hellenistic period that human beings naturally distinguished between what is good and what is evil, and this is reflected in the usage of *syneidesis* in *Romans* 2:14–15 (Marietta 1970, pp. 176–87). Even until Paul's letters, conscience primarily referred to a kind of moral awareness of good or evil actions (Pierce 1955; Bosman 2003). The suppression of Christianity in the Roman Empire in the first three centuries CE, and subsequent attacks by the Catholic Church on heretics, resulted in an expansion of the meaning of conscience. It came to include one's ability or rights to determine one's own beliefs freely, as in the phrase "freedom of conscience". This double meaning of conscience continues today (Chadwick 2006; Sorabji 2014; Wang 2023, p. 154).

In the first half of his article, Stendahl argued that Paul's doctrine of justification by faith was primarily intended to remove obstacles to the entry of Gentiles into God's covenant with the Jews. The doctrine also provided an answer to the question of whether Gentile Christians would usurp the Jews' special place in the process of salvation: the law applied only to the Jews and its purpose had been fulfilled in the coming of Christ (*Gal.* 4:23). As long as the Gentiles renounced the literal requirements of the law and believed in Christ instead, they would be called righteous, becoming the children of God and the heirs of the inheritance of Abraham (*Rom.* 8:17; *Gal.* 3:29).

In Stendahl's view, Paul's doctrine of justification by faith only aimed at dealing with the problem of Gentile Christians in the context of the first century. It did not aim to provide a general solution to individuals' inner conflicts in the course of their faith, which

Augustine and Luther interpreted in Paul's texts. However, Stendahl neither elaborated on these complex hermeneutic controversies nor comprehensively studied the different uses of conscience in Paul's letters. His assessment was limited to Paul's statements about his own conscience, which was a "robust conscience". For example, Paul confirmed that he always fulfilled all requirements of the law and was, "as to righteousness under the law, blameless" (*kata dikaiosynēn tēn en nomō genomenos amemptos*, *Phil*. 3:6) before his conversion on the road to Damascus; after the conversion, he never doubted his apostleship (*Gal*. 1:12) nor his ability to abide by God's commandments, and never suspected the physical illness of his body to be a sign of his own sin (*Gal*. 4:13; 1 *Cor*. 12:7–8). In contrast to Augustine and Luther, Paul himself does not appear to have experienced a plagued conscience (1 *Cor*. 4:4; 2 *Cor*. 2:12; *Acts* 23:1; Stendahl 1963, pp. 206–11).

Any reckoning of the status of Paul's conscience, and its disturbance or tranquility in the face of sin, must contend with disputes around *Romans* 7:14–25. Throughout the exegetical history of *Romans* 7, many debates have focused on this passage (Fitzmyer 1993, pp. 472–77; Jewett 2007, pp. 461–67). Opinion differs as to whether this passage is autobiographical, and varied interpretations of the reference intended by "I" exist, specifically whether it includes or excludes Christians alongside unbelievers. Questions have been raised around the passage as a description of Paul himself, and whether it might describe Paul before or after his conversion. Some consider that the conflict described between "I" and the flesh shows the conscience of the referent of "I" being disturbed by sin.

Stendahl's interpretation of this passage contains a number of distinctive points. Firstly, Paul's sin was the persecution of the church; he did not sin again after conversion, and never suffered the disturbance of a "plagued conscience". Secondly, Stendahl reads *Romans* 7 as an explanation of the law in order to justify its holiness and goodness; in the passage, "I" is not identical to sin or flesh. Finally, *Romans* 7:19 does not directly lead to the lament in 7:24, rather Stendahl considers that 7:20 and subsequent verses distinguish between "I" on one side, and, on the other, both the evil that "lies close at hand" (*emoi to kakon parakeitai*, *Rom.* 7:21) and "this body of death" (*Rom.* 7:24). Stendahl thus argues that "I" is not involved in sin and there is no division of the self.

As to this exegetical history, Stendahl commented that "this Western interpretation reaches its climax when it appears that even, or especially, the will of man is the center of depravation" (Stendahl 1963, p. 213). It is evident that this climax refers to Augustine's philosophy of will. In the last decades, this well-known paper of Stendahl paved the way for a hermeneutic movement called "New Perspective of Paul" (Sanders 1977; Chen 2006; Zhang 2011; Kässmann 1971, pp. 60–78; Farnell 2005, pp. 189–243; Valčo 2012, pp. 206–30; Maxwell 2013, p. 145). Later interpreters have treated this passage as a general exposition of sin as it relates to human nature, marginalizing Paul's focus on the law.

The "New Perspective of Paul" was centered on reinterpretation of Paul's letters and the rejection of Lutheran theology, but lacked an in-depth examination of the interpretative approach and theoretical foundations of Augustine. In view of this, the second section of my paper argues that Augustine's idea of introspective conscience contrasts with the usages of conscience in ancient Greek literature and in Paul's letters. This contrast arose as a logical consequence of Augustine's philosophy of will.

## 3. Augustine: Will, Grace, and the Discovery of Introspective Conscience

Augustine inherited the concept of conscience from the Latin tradition. His uses of the term *conscientia*, which appears at least ten times in *Confessions*, are consistent with this tradition (*Confessions* 1.18.29, 2.5.11, 4.9.14, 5.6.11, 8.7.18, 10.2.2, 10.3.4, 10.6.8, 10.30.41, 12.18.27). The only passage in which Augustine's *conscientia* exceeds the scope of moral cognition is found in *Confessions* 8.7.18, where *conscientia* questions why Augustine has not decided to believe in Christ. However, as Stendahl notes, Augustine enriched and deepened the meaning of *conscientia* by discovering the concept of introspective conscience. Understanding this discovery requires an analysis of the basic structure of Augustine's phi-

losophy of will. This analysis clarifies why, within Augustine's philosophy of will, God's grace cannot offer complete comfort to an introspective conscience.

Augustine's philosophy of will sits within a divinely created order: the all-good God created a good world, in which evil is not a physical entity but a "privation of good" (*privatio boni*), as argued in the refutation of Manichaeism found in *De libero arbitrio*. In this created order, living things are organized within a hierarchy, with the human soul, alongside the souls of all the angels, occupying the highest level. The human soul has its own higher and lower parts, in which the higher parts consist of intellect, will, and memory. Among these, will serves as a "middle good" (*medium bonum*), in that it can direct itself toward God, the soul, or other creatures. The latter two choices constitute the fall of the will and the origin of evil. After the first sin, the will falls further into ignorance (*ignorantia*) and difficulty (*difficultas*), so that human beings eventually become incapable of doing good and instead fall necessarily into doing evil. Augustine insisted that although doing evil is out of necessity, it is also out of free will, and human beings must, therefore, bear moral responsibility for it. The corruption of the will undermines the "image" and "likeness" created in human beings by God at the beginning (*Gen.* 1:26), and causes the soul's higher parts to lose control over the body. The body's rebellion and disturbance degrade human beings from a state of "being able not to sin" (*potest non peccare*) to a state of "being unable not to sin" (*non potest non peccare*).

Augustine's division of a person into soul and body, and the identification of the soul as the essential part of a person, is influenced by Neoplatonism. Only by turning from the external material substance to the internal spiritual substance (*interius cogitando*), from sensory to intellectual stimuli, and from the soul's lower parts to its higher parts, can the soul gradually ascend to the divine realm. This ascension enables the soul to glimpse God, who is wisdom itself (*Confessions* 9.10.24). Based on this theological anthropology, Augustine discovered the idea of an "inner self" (*interiore homine*) that is the subject of corruption, sin, justification, sanctification, and introspective conscience. Since the will is totally corrupted and always bound by sin, human beings have lost any possibility of self-redemption, but only continue to commit evil and sin.

In his commentaries on *Romans* in the mid-390s, Augustine divided the developmental history of all human individuals into four stages, "before the law" (*ante legem*), "under the law" (*sub lege*), "under grace" (*sub gratia*), and "in peace" (*in pace*), following the sequence of law and grace given by God (*Expositio quarundam ex epistula apostoli ad Romanos* 13–18.2). When the law has been given but grace has not, human beings are "under the law". In this state, the function of the law is not to save, but rather to reveal sin and provide "the knowledge of sin" (*Rom.* 3:20).

The soul's fall begins with the fall of human will, whereas its salvation begins with God's grace as a gift. Questions around the priority of God's grace over human will at the "beginning of faith" (*initium fidei*) mark an important area of development in Augustine's commentaries on *Romans*. In the first commentary, Augustine defended the priority of human will at the beginning of faith based on its autonomy: human will cannot be forced to believe, but must make an active choice to believe. Interpreting *Romans* 7:14–24, Augustine argues that "I" refers to people "under the law", meaning that the people described have not received grace and that Paul was speaking on behalf of these non-Christians. If this is the case, however, Augustine would have to explain why these people can still "will what is right" (*bonum velle*) before receiving grace (*Rom.* 7:18–19; *Expositio quarundam propositionum ex epistula apostoli ad Romanos* 44.1–3). Augustine comments further on this issue when considering the election of Jacob and the non-election of Esau in *Romans* 9. Here, he makes it clear that some sinners are able to will good independently, so that God, who recognizes these "most secret merits" (*occultissimus meritis*), identifies them as righteous, grants them grace, and punishes other sinners (*De diversis quaestionibus octoginta tribus* 68.4). Thus, in his first three commentaries on *Romans*, Augustine is very close to claiming that humans remain capable of willing good and believing in God independently, even when the soul is in its fallen state.



Augustine's thinking on this issue developed over time. In book three of *De libero arbitrio*, Augustine suggests that the corrupted will fails disastrously in its ignorance and difficulty, a position which is obviously in conflict with his conclusions in the commentaries on *Romans*. Further interpretations of passages from *Romans* appear in Augustine's *Ad Simplicianum*, a collection of responses to exegetical questions raised by Simplicianus, written in 396. The first book of these responses shows that Augustine took contrasting approaches to reinterpreting *Romans* 7 and 9.

Regarding *Romans* 7, Augustine retained the main argument of his previous commentaries, minimizing the power of divine grace just enough to create a role for the individual's will at the beginning of the individual's faith. However, Augustine offered a more literal explanation of the election of Jacob in *Romans* 9. Augustine holds that God elected Jacob before he was born, and infers from this that God's grace precedes and promotes the good work of human beings. He further surmises that God could not elect Jacob on the basis of his good work, because he had not been born yet. Finally, Augustine notes that God could not have elected Jacob based on the foreknowledge that he would believe in God since this would entail foreseeing his good work.

Augustine concludes that it is not possible to distinguish God's basis for the election of Jacob. Eventually, Augustine came to abandon the idea of "the most secret merit", and instead aligned his philosophy of will more closely with *Philippians* 2:12–13, where even the "good will" (*bona voluntas*) of individual people arises from God's work in their hearts, and this grace initiates a person's faith in God before any work of their own will. Augustine abandons explanations for God's election of Jacob but not Esau, appealing to God's "most secret justice that is far removed from human understanding" (*aequitate occultissima et ab humanis sensibus remotissima*) (*Ad Simplicianum* 1.2.16, Boniface Ramsey's translation), and acknowledging that human beings cannot comprehend this justice. In *Retractationes* 2.1.3, Augustine reflected on this development in his philosophy of will: "I in fact strove on behalf of the free choice of the human will, but God's grace conquered" (Augustine 2010, p. 110).

In order to fully understand Augustine's discovery of the introspective or plagued conscience (*syneidesis*), we can now summarize Thomas Aquinas's reinterpretation of the same subject. Following Jerome (ca. 347–419/420) and Bonaventure (1221–1274), Aquinas (ca. 1225–1274), with other theologians in the Middle Ages, differentiated conscience (*conscientia*) from the spark of conscience (*synderesis*). In his works, *synderesis*, a new Latin word, was used simultaneously with *conscientia*. For Aquinas, as Sorabji analyzed, *synderesis* is "a disposition (*habitus*) within the power of reason", which is never fallible, but meanwhile, conscience (*conscientia*) is fallible and "can make mistakes" (Sorabji 2014, pp. 59–67; Wang 2023, pp. 154–56). We will see that Augustine, based on the total corruption of human will and the necessity of divine grace, would never endorse Aquinas's new interpretation of conscience.

## 4. Augustine on Conscience: Before and after Conversion

Augustine's exegesis of *Romans* led him to divide the development of a person into three stages: a first stage without grace, a second stage with grace and before conversion, and a final stage with grace following conversion. For Augustine, only those people in the last two stages are affected by a "plagued conscience".

As Stendahl argued, Paul had a "robust conscience" both before and after his conversion. As a Pharisee, Paul was "blameless" (*amemptos*) in the law (*Phil*. 3:6); when he was an apostle to the Gentiles, he became "all things to all people" (*1 Corin*. 9:22). In both situations, Paul's conscience was never disturbed by sin, but was always full of peace and confidence. Augustine, by contrast, received such peace and confidence neither before nor after his conversion. In *Confessions*, he recalled that his spiritual journey before conversion was far "from your unmoved stability" (*ab stabilitate tua*), and even remarked that "I became to myself a region of destitution" (*et factus sum mihi regio egestatis*, *Confessions* 2.10.18, Augustine 1991, p. 34). Describing his own conversion in books 7 and 8, Augustine also lamented the stubborn habits that presented serious impediments to his moral conversion,

whereas his intellectual conversion to the Catholic faith was not blocked by these habits (O'Meara 2001, p. 125; Fredriksen 1986, pp. 3–34).

Augustine emphasized the challenges presented by his "plagued conscience" throughout his own conversion. He affirmed that divine grace constantly guided him to discard pagan and heretical teachings, helping to eliminate all worldly desires. While Augustine in *Ad Simplicianum* described this grace as an external call that opens the door of faith for human will, the grace he described in *Confessions* became an inner grace. Augustine not only recognized the presence of grace in his life before conversion but also admitted that it operated at all times, even in his childhood. His conversion experience in the garden of Milan in 386 exemplifies this. Augustine imagined a long process of conversion, which ultimately manifests as the final decision of the will over the issue of faith. Through this decision, it becomes evident that Augustine's "plagued conscience" is always accompanied by operative divine grace.

Augustine's own conversion exemplifies the way in which God's grace directs human will toward faith. When this good will constrains other human desires, the soul enters a state of fierce internal conflict. In *De libero arbitrio* 1.12.26 and 3.3.7, Augustine affirms several times that human will is completely within our power to start with, and cannot be externally compelled or hindered. This original will, as well as the corrupted will that has not been redeemed by grace, is not internally divided. However, when grace directs the human will toward faith, the will is split between good and evil, which gives rise to an internal conflict. Augustine vividly describes this inner conflict:

> "I sighed after such freedom, but was bound not by an iron imposed by anyone else but by the iron of my own choice. The enemy had a grip on my will and so made a chain for me to hold me a prisoner. The consequence of a distorted will is passion. By servitude to passion, habit is formed, and habit to which there is no resistance becomes necessity. By these links, as it were, connected one to another (hence my term a chain), a harsh bondage held me under restraint. The new will, which was beginning to be within me a will to serve you freely and to enjoy you, God, the only sure source of pleasure, was not yet strong enough to conquer my older will, which had the strength of old habit. So my two wills, one old, the other new, one carnal, the other spiritual, were in conflict with one another, and their discord robbed my soul of all concentration". (*Confessions* 8.5.10, Augustine 1991, p. 140).

For Augustine, the will, that was initially created good has become evil by accumulating habits that are able to resist the good will that arises through God's grace. This results in an internal conflict between good and evil wills in the soul: "so I was in conflict with myself and was dissociated from myself" (*Confessions* 8.10.22, Chadwick's translation, p. 148). This inner division or conflict is a manifestation of the introspective conscience, which goes beyond moral judgment in the ancient Greek and Hellenistic senses to involve moral self-criticism and confession of one's inescapable guilt.

For Augustine, this inner conflict led him to deviate from the moral principles set out for him by his mother, Monica, and his teacher, Ambrose. On his long journey to conversion, Augustine craved wealth, social status, and marriage, and even took a lover. After grace is given, the evil will resists the new good will, and the converted person becomes increasingly aware of the tight bonds of their own sin, which can only be broken by the operation of God's grace. Augustine chose during his conversion to face God and pray to Him alone, confessing his affliction by sin: "For I felt my past to have a grip on me" (*Confessions* 8.12.28, Augustine 1991, p. 152). Although God's grace initiates human faith, as shown through Augustine's private prayer, it does not eliminate sin.

Sin and guilt from the past disturb the introspective conscience, and cannot be escaped, before or after conversion. Although the "plagued conscience" described in *Confessions* 8.9.21–8.12.29 can be understood as a pre-conversion psychological phenomenon, which does not affect Christians after conversion, the process of sanctification suggests that inner conflict persists after conversion. On hearing a voice singing, "Pick up and read,

pick up and read" (*tole lege, tole lege, Confessions* 8.12.28, Augustine 1991, p. 152), Augustine picked up Paul's letters and, on reading *Romans* 13:13–14, "it was as if a light of relief from all anxiety flooded into my heart. All the shadows of doubt were dispelled" (*Confessions* 8.12.29, Augustine 1991, p. 153). Here the light represents God's operative grace, by which Augustine eventually completed his conversion. However, all Christians begin a process of sanctification after conversion. This process starts when a person is first called righteous by God and ends when their life on Earth comes to an end. Death is not final, but only marks the beginning of judgment that will evaluate a person's journey from justification to sanctification. Throughout their journeys, despite the help of God's operative grace, human beings are disturbed by sin and the weakness of will.

Augustine's early commentaries on *Romans* 7 emphatically argue that the "I" of that chapter does not include the Christian Paul. Rather, he was using the voice of non-Christians (De Bruyn 1993, p. 105). However, Augustine changed his position in his debates with Pelagius (*Retractationes* 1.23.1). In *De peccatorum meritis et remissione et de baptismo parvulorum* 1.27.43 and 2.12.17, Augustine concedes for the first time that people who have converted and received divine grace can also be subject to concupiscence (*Rom*. 7:22–25). This change took place in 421 AD with the completion of *Contra duas epistulas Pelagianorum*, in which Augustine held that the "I" in *Romans* 7 includes people under grace; it would even be possible for the apostle Paul to fall into the dilemma of doing evil while willing good, and the exclamation "wretched man that I am!" (*talaipōros egō anthrōpos, Rom*. 7:24) could apply to him as to any person. Augustine acknowledges Paul's identity as a saint, at the same time asserting that he also suffered from "plagued conscience" throughout his debate with Julian of Eclanum, a Pelagian bishop. This assertion indicates Augustine's opposition to moral perfectionism and faith elitism, positions that the Pelagians endorsed. Augustine's position finds a complement in "the righteousness of the Law" (*nomon dikaiosynēs, Rom*. 9:31), which Paul adhered to before his conversion (*Sermons* 169.3). Luther, following this Augustinian interpretation, naturally regarded the "I" as a post-conversion Christian and thus proposed the idea that a believer is "righteous and sinful simultaneously" (simul iustus et peccator; Reasoner 2005, pp. 75–76).

Augustine insists in the later works that baptism after conversion removes the original sin inherited from ancestors, but cannot wash away concupiscence as its residue. He uses this to explain how people after conversion can still commit the sin of greed, and to justify the petition to "forgive us our debts" (*aphes hēmin ta opheilēmata, Matt*. 6:12) found in the Lord's prayer. In *Confessions* 10.35.57, Augustine also states that he continued to be distracted by surrounding events, such as the hunt occurring in the countryside or the lizard catching flies during his theological reflection, even after becoming a bishop. Particular vulnerability to these distractions arises through dreams, which even the saints cannot avoid, although they can be free from God's punishment. In *Contra Julianum* 4.2.10, Augustine states that:

> "And if it should steal from them the slightest consent, even in sleep, they groan when they have weakened and say, 'How my soul is filled with illusions' (*Ps*. 38:8). For when dreams deceive the sleeping senses, even chaste souls somehow fall into such shameful consent. If the Most High held such consent against them, who would live in chastity?" (Augustine 1998, p. 385)

Augustine is open to the possibility that all Christians, including the apostle Paul, may suffer weakness of will in the process of sanctification. Anyone might commit sins in involuntary dreams, and both the advent of grace and the temptation of sin may be found in them, sometimes simultaneously.

Based on foreknowledge (*proegnō, praescere*) and predestination (*proōrisen, praedestinare*), two words used by Paul in *Romans* 8, Augustine developed a stronger theory of predestination in his later works. Unlike the double predestination of Calvinism, Augustine's theory was of the single predestination, in which God actively predetermines the salvation of some people (e.g., Jacob) and helps them become saints with his grace (Burns 1980). While some texts imply that certain individuals are "predetermined to eter-

nal death" (*quos praedestinavit ad aeternam mortem*, *de anima et eius origine* 4.16), Augustine interprets this as those individuals' continuation in sin, passively allowed by God, followed by their eventual punishment with eternal death. Eternal death for these individuals is, therefore, not predetermined by God (Zhou 2009, pp. 227–29). Unlike the supralapsarianism of Calvinism, which sees individual sin as a product of God's predestination, Augustine saw an individual's sin as a result of free will (Bonner 2007, pp. 45–46).

Since human will is corrupted, it is only out of God's active agency that human beings receive grace, salvation, and predestination, and individual people are only passive recipients. In this view, the process of salvation belongs to God alone and is separate from any human influence. This pulls any balance between a person's ethical self-assertion and their religious self-surrender apart from the process of salvation. As a result, human beings are almost totally deprived of the opportunity to bring their own salvation about through self-assertion and self-confirmation of their morality (Wetzel 2000, pp. 123–24). Under God's absolute sovereignty, human beings become unable to give effect to their own good wills or deeds, which arise entirely as a result of God's grace. No individual can be certain of predestined redemption, since there is no difference discernible to living people between those who will be redeemed and those who will not. Augustine's philosophy of will, coupled with his theory of predestination, creates conditions in which the introspective conscience of any Christian, even after conversion, is still plagued. No person can confirm their own redemption but must suffer an inner torture of uncertainty until the judgment day.

Faced with this uncertainty, those who have not converted will certainly be convicted, and those who are called to be righteous must always be alert and cautious. The latter must express their faith by constantly repenting their sins and praising God, and justify this faith by living out the two commands of loving God and loving one's neighbors as oneself. The ultimate authority of judgment is left to God's "most secret justice". From justification to sanctification, an individual conscience is held in suspense and surrounded by uncertainty under God's predestination. For Augustine, leaving the sovereignty of redemption to God allows human beings to focus on the practice of their faith. At the same time, Augustine compares God's redemptive sovereignty with "snares of truth" (*retia veritatis*), in which attempts to escape are akin to suicide, leading only to a death like hurling oneself from a precipice (*se abrupta praecipitent*, *Epistulae* 194.8.35).

Augustine does not make the effectiveness of operative grace explicit and discusses neither the extent of its power to overcome sin nor any limitations to its effectiveness. Ambiguous imagery, metaphor, and unanswered questions are commonplace in biblical exegesis. Paul, responding to questions about whether God's election of Jacob is just in *Romans* 9:19–23, invoked the metaphor of pottery to emphasize the differences or absolute distance between the creator God and His human creations. God gave His own response to Job's question about why a righteous man should suffer in a whirlwind and responded with another question: "Where were you when I laid the foundation of the earth? Tell me, if you have understanding" (*Job* 38:4). For Augustine, it is likely that faith and praise of God were prerequisites to resolving conflicts between theoretical arguments, as the intermingling of his incisive questions with his prayers in *Confessions* suggests.

## 5. Mencius: A Robust and Extensive Conscience

In pre-Qin literature, Mencius was the first to use the word conscience (*liangzhi*, 良知). It appears only once in *Mencius* 7A15, where it denotes a kind of knowledge that is "known without having to think". Mencius also affirmed that all people possess this knowledge from childhood:

> 孟子曰："人之所不学而能者，其良能也；所不虑而知者，其良知也。孩提之童无不知爱其亲者，及其长也，无不知敬其兄也。亲亲，仁也；敬长，义也；无他，达之天下也。"(7A15)

Mengzi said, "What people are able to do without having learned it is an expression of original, good ability (*liangneng*, 良能). What they know without having to think about it is an expression of original, good knowledge (*liangzhi*, 良知). There are no young children

who do not know enough to love their parents, and there are none who, as they grow older, do not know enough to respect their older brothers. To be affectionate toward those close to one—this is humaneness (*ren*, 仁). To have respect for elders—this is rightness (*yi*, 义). All that remains is to extend these to the entire world" (7A15, Bloom 2009, p. 147).

Several translators have given almost the same English translation of *liangzhi*, including "intuitive knowledge" (Legge 1992, p. 519), "innate knowledge" (Chan 1963, p. 80), "genuine knowledge" (Van Norden 2009, p. 80), and "original, good knowledge" (Bloom 2009, p. 147). In this paper, I translate *liangzhi* as conscience in order to help us compare Mencius's view with Paul's and Augustine's views.

There are different interpretations of *liang* (良) here, such as "even" (甚, Zhao Qi and Jiao Xun) or "natural goodness" (本然之善, Zhu Xi) (Shun 1997, p. 188). In the commentary on *Mencius*, Zhu Xi quotes Cheng Hao's (程颢, 1032–1085) interpretation, "both intuitive knowledge and inherent ability are endowed by Heaven, and are neither derivative from any external cause nor dependent upon any human input" (Wang Keyou's translation with a small correction, Guo 2017, p. 51; Zhu 1983, p. 353). But as Van Norden has argued, in this text, the "knowledge" (*zhi*) here is definitely good; as a result, "good" (*liang*) is not a useful indicator of the nature of the knowledge in question, but instead serves to emphasize its innate source. For him, conscience cannot be translated as "best knowledge", but only as "genuine knowledge": "'genuine' marks the contrast between what is 'ingenuous' or 'original' as opposed to what is 'artificial' or 'acquired'" (Van Norden 2009, p. 126). Alternatively, Yang Bojun objects to translate it into other Chinese words. "It is much more appropriate not to translate this specific philosophical term of Mencius" (Yang 2018, p. 342).

*Mencius* 7A15 affirms that every person is born with original ability and conscience. The examples given are that all children know from an early age to love their parents, and when they grow up, they know to respect their brothers. Familial affection and regard for one's elders produce the virtues of humaneness and rightness, respectively. In these examples, conscience naturally produces the original ability, so that knowledge and ability effectively become two sides of the same coin. The commentary of Zhao Qi reinforces this view: "conscience is also the ability" (Zhao and Sun 1999, p. 359). Conscience is a kind of innate moral knowledge, distinct from acquired knowledge of external objects, which concerns a direct, non-empirical grasp of what is good and what is evil. This is complemented by a direct, non-empirical identification with moral principles.

It has been observed, however, that conscience in *Mencius* 7A15 is still in its infancy and not fully mature (Liang 2008, p. 344). Van Norden objected to Wang Yangming's interpretation of conscience:

> "The later Confucian Wang Yangming emphasized the phrase 'genuine knowledge'. However, for him as for other School of the Way philosophers, it does not refer to an incipient tendency but rather a fully developed faculty of simultaneous ethical insight and motivation". (Van Norden 2009, p. 127; Ivanhoe 2002, pp. 48–50)

Specifically, in the early concept appearing in *Mencius* 7A15, conscience first cultivates the virtue of humaneness by loving one's parents, then cultivates the virtue of rightness by respecting one's elder brothers. After these developments, conscience continues to mature and progressively extends these virtues from families to compatriots, and from a country to the whole world.

Although the word conscience appears only once in *Mencius*, it is often linked to Mencius's four minds (*sixin*, 四心) argument and the four sprouts of virtue (*siduan*, 四端). The four sprouts of virtue tend to be regarded as an idea that follows from the four minds argument. Together, they form Mencius's overarching argument for the goodness of human nature.

The four sprouts of virtue appear in an example Mencius uses to argue that "all human beings have a mind that commiserates with others" (*ren jieyou burenren zhixin*, 人皆有不忍人之心): "if anyone were suddenly to see a child about to fall into a well, his mind would be filled with alarm, distress, pity and compassion" (2A6, Bloom 2009, p. 35).

"恻隐之心，仁之端也；羞恶之心，义之端也；辞让之心，礼之端也；是非之心，智之端也。人之有是四端也，犹其有四体也。" (2A6)

"The mind's feeling of pity and compassion is the sprout of humaneness; the mind's feeling of shame and aversion is the sprout of rightness; the mind's feeling of modesty and compliance is the sprout of propriety (*li* 礼); and the mind's sense of right and wrong is the sprout of wisdom (*zhi* 智). Human beings have these four sprouts just as they have four limbs". (2A6, Bloom 2009, p. 35)

In the debate with Gaozi (告子, ca. 420–350BC) about human nature, Mencius insisted that human nature is not neutral or evil, but totally good (Yang 2015, pp. 44–52). One analogy used to support this contention was that "the goodness of human nature is like the downward course of water" (6A2, Bloom 2009, p. 121; Xu 2019, pp. 37–48). Continuous cultivation of the sprouts of virtue can regulate families (*qijia*, 齐家), bring order to the country (*zhiguo*, 治国), and bring peace to the world (*pingtianxia*, 平天下). At 6A2, Mencius also makes use of an analogy of a person's four sprouts of virtue, comparing them to a person's four limbs (Li 2021). Conscience can be equated with the four feelings of the mind and the four sprouts of virtue. In each situation, the mind's feelings are manifestations of conscience.

Among the mind's four feelings, however, conscience seems much closer to the sense of right and wrong (*shifei zhixin*, 是非之心). It is this knowledge by which a person knows what is morally right or wrong, good or evil. In his doctrine of four axioms (*sijujiao*, 四句教), Wang Yangming also gave the same definition. "The faculty of innate knowledge is to know good and evil" (Chan 1963, p. 688). This may be one of the reasons why *syneidesis* in *Romans* 2:15 is translated as "是非之心" in the Chinese Union version of the Holy Bible (Zhang 2015, pp. 76–77). Tang Junyi comments directly on the relationship between conscience and knowledge in Neo-Confucianism: "Cheng Yi's keeping the Way and eliminating desires and Wang Yangming's extension of conscience, strictly speaking, are all based on Mencius's mind which has the sense of right and wrong" (Tang 1984, p. 76).

In Mencius's view, human beings are born with conscience by which they can engage in the cultivation of their own virtues. On comparing this view with those of Paul and Augustine, a number of key insights into Mencius's theory of conscience become clear, as well as some dangerous deficiencies.

Firstly, for Mencius as for Paul and Augustine, the primary functions of conscience are to distinguish good from evil and to reflect on the good or evil qualities of a person's own moral behavior. Mencius also believed that conscience is innate. Everyone has an understanding of right and wrong from birth, whether it is granted by Heaven or the Christian God.

Secondly, Mencius displays great confidence in his own virtues and moral actions, in a manner similar to the "robust conscience" of Paul. Parts of the text, in which Mencius appears to conduct self-evaluation during an intimate conversation with his disciples, indicate that he may have believed in his own role as a savior:

"我四十不动心。" (2A2)

"Since the age of forty my mind has been unmoved". (Bloom 2009, p. 29)

"我知言，我善养吾浩然之气。" (2A2)

"I understand words. I am good at nourishing my vast, flowing *qi*". (Bloom 2009, p. 30)

"夫天未欲平治天下也；如欲平治天下，当今之世，舍我其谁也？" (2B13)

"Heaven does not yet want to bring peace to the world. If it wanted to bring peace to the world, who is there in the present age apart from me?" (Bloom 2009, p. 48)

Just as Paul wrote of his own adherence to the law and declared that he had received and was preaching "the truth of the gospel" (*tēn alētheian tou euangeliou*, *Gal.* 2:14), Mencius both stated his belief that he had recognized the Way of Heaven and was practicing it, and

also announced his intention to save the corrupt and war-torn world through his Confucian wisdom and writings.

Thirdly, unlike Augustine's "introspective conscience" or "plagued conscience", Mencius's conscience, after recognizing good and evil, actively does good and eliminates evil of its own volition. Mencius's conscience constantly corrects moral motivations, and as such is the key driver of subsequent moral actions:

"万物皆备于我矣。反身而诚，乐莫大焉。强恕而行，求仁莫近焉。" (7A4)

"All the ten thousand things are complete in me. To turn within to examine oneself and find that one is sincere—there is no greater joy than this. To dedicate oneself in all earnestness to reciprocity—there can be no closer approach to humaneness". (Bloom 2009, p. 144)

This inner satisfaction explains why Mencius does not lament as Paul did in *Romans* 7 that "I can will what is right, but I cannot do it" (*to gar thelein parakeitai moi, to de katergazesthai to kalon ou*, *Rom.* 7:18), nor does he perceive division and weakness of his own will. Mencius does not appear vulnerable to the necessity of doing evil before his conversion to faith and is not concerned with temptation by carnal desires after conversion. There is no need for repentance of sin, and no imperative to pray for the intervention of God's operative grace, both of which Augustine felt keenly in *Confessions* and later works.

Fourthly, unlike Paul and Augustine who always emphasized the necessity of grace from God, Mencius has an assured confidence that conscience, even in its earliest stages of developing the sprouts of virtue, is already self-sufficient. For Mencius, a human being who reflects upon their inner self and expands their own conscience will make progress from serving their parents to protecting "all within the four seas", and finally to realizing the Way of Heaven. As Tao Jiang argues, this normative Mencius is a ren-based extensionist (Jiang 2021, pp. 157–60). In this process, there is no one who needs additional help beyond what their conscience can provide. In the Christian context, we can say that Mencius would be a follower of Pelagius (ca. 354–418) and, just like Peter Brown's comment on Augustine's position in the first two books of *De libero arbitrio*, Mencius is also "more Pelagian than Pelagius" (Brown 2000, p. 141).

"凡有四端于我者，知皆扩而充之矣，若火之始然，泉之始达。苟能充之，足以保四海；苟不充之，不足以事父母。" (2A6)

"When we know how to enlarge and bring to fulfillment these four sprouts that are within us, it will be like a fire beginning to burn or a spring finding an outlet. If one is able to bring them to fulfillment, they will be sufficient to enable him to protect 'all within the four seas'; if one is not, they will be insufficient even to enable him to serve his parents". (Bloom 2009, pp. 35–36)

Mencius's conscience appears to be a synthesis of Paul's "robust conscience" with Augustine's "operative grace", forming a kind of "operative conscience". This conscience is not subject to the travails of Augustine's "plagued conscience", and is capable of realizing outer kingship (*waiwang*, 外王) by its own self-sufficient inner sagehood (*neisheng*, 内圣). The most significant deficiency, however, lies in the optimism inherent in Mencius's argument for the goodness of human nature and the extensive capabilities of conscience. Other texts suggest a more pessimistic view. Mencius also wrote: "that wherein human beings differ from the birds and beasts is but slight. The majority of people relinquish this, while the noble person retains it" (人之所以异于禽兽者几希，庶民去之，君子存之, 4B19, Bloom 2009, p. 89). However, this pessimistic social reality is not further explored in surviving texts, nor does Mencius explain what it means for the majority of people to ignore this slight difference. Mencius fails to examine the fragility of good human nature and the weakness of the human will to do good as discussed in Plato's (*Republic* 439e–440a) in the Western philosophical tradition (Dahl 1984; Gosling 1990). His confidence in an "operative conscience" simply disregards the possibility that disruptive physical desires might impede the continuous expansion of a person's conscience.

## 6. Conclusions

Augustine's commentaries on *Romans* shifted the core issue of Pauline theology from moral cognition between good and evil to moral criticism of one's guilt and sinfulness, and thereby introduced the subject of introspective conscience for the first time in the history of Western thought. Augustine developed the idea of a pre-conversion "plagued conscience" that differed markedly from Paul's "robust conscience". The theoretical foundation for this idea, upon the basis of his doctrine of will and grace, further supported a persistent conflict between good and evil desires in one's inner self. A post-conversion "plagued conscience" is also discernible on the basis of Augustine's doctrine of predestination. Since the self-recognition of faith does not guarantee salvation, one's inner self continues to be affected by the certainty of one's faith and the uncertainty of God's salvation. This feature of Augustinian introspective conscience bolstered monastic movements and asceticism in the fourth and fifth centuries and was adduced in support of pietism and mysticism in the Middle Ages. Luther, as a monk in an Augustinian monastery, carefully studied Paul's doctrine of justification by faith in *Romans*. Augustine's influence encouraged Luther to seek salvation through human faith and God's faithfulness, relieving the torture of introspective conscience. Ironically, the doctrinal legacy Luther inherited from Augustine contributed toward the split of the Catholic Church, whose unity Augustine had defended for all his life.

Unlike Paul and Augustine, Mencius believed that conscience is self-sufficient even in its earliest stages of encouraging the sprouts of virtue. It does not need help through God's grace or the power of Heaven. For Mencius, the constant expansion of a person's conscience achieves both self-cultivation and the correction of the distorted world. However, Zhang Hao criticized Mencius's self-expanding and self-saving Confucian conscience as excessively optimistic in that it did not fully recognize the weakness of dark consciousness (*youan yishi*, 幽暗意识) in human nature. This shortcoming in Mecius's understanding of conscience resulted in a lack of attention to the external discipline required to restrain physical desires and to the possibility that conscience itself might become corrupted. In his view, the expansion of conscience from inner sagehood to outer kingship argued by Mencius was never realized in ancient China and it did not initiate a democratic tradition in modern China; in contrast, this optimistic Confucian argument often became a vassal or accomplice of the autocratic monarchy (Zhang 2018, pp. 43–58; Shen 2011, pp. 133–44).

**Funding:** This research is funded by The National Social Science Fund of China, "Augustine's Complete Letters: Research and Translation" (21BZJ009).

**Institutional Review Board Statement:** Not applicable.

**Informed Consent Statement:** Not applicable.

**Data Availability Statement:** Not applicable.

**Conflicts of Interest:** The author declares no conflict of interest.

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
