# Peer review of "The Dilemma of Conscience: From Paul and Augustine to Mencius"

_religions, doi:10.3390/rel15030265_

Round 1
Reviewer 1 Report
Comments and Suggestions for Authors
This manuscript is well-crafted and engaging, meriting recommendation for publication. It effectively juxtaposes Western Christian (Paul, Augustine) and Chinese (Mencius) conceptions of conscience, illustrating notable parallels.
However, several areas could benefit from refinement:
(1) Introduction: The statement, ‘In both Chinese and Western moral philosophy, the concept of conscience has a long tradition’, suggests a similarity in the conceptualisation of conscience across these traditions. To clarify this assertion, it would be beneficial to present the principal definitions of conscience from both Western and Chinese perspectives in the Introduction, thereby facilitating a deeper understanding of these similarities.
(2) The predominance of Luther in the author’s analysis of Augustine’s interpretation raises questions, given that Western Christian theological thought has been arguably more influenced by Thomas Aquinas. Mentioning Aquinas’s reinterpretation of Augustine on the matter of conscience would provide a more comprehensive understanding of the theological discourse. Furthermore, while the author’s reference to Luther is valid, acknowledging Aquinas’s contributions would enrich the discussion.
(3) The author references Krister Stendahl’s significant work, ‘The Apostle Paul and the Introspective Conscience of the West’; nevertheless, Stendahl was not a major European theologian and is not widely known. Thus, it would be worth pointing out his theological and denominational background and the basis of his interpretation of Paul and Augustine. This context would enhance readers’ appreciation of Stendahl’s contribution.
(4) Technical issue: The author’s approach to including Chinese quotations, complete with characters and often Latin transliteration, juxtaposes with the treatment of Greek terms, which are only provided in Latin transliteration. Incorporating original Greek notation alongside the transliteration would offer additional value to the article, enriching the textual analysis and providing a more authentic representation of the sources.
Reviewer 2 Report
Comments and Suggestions for Authors
The Dilemma of Conscience: From Paul and Augustine to Mencius
1. This article is a very interesting one. However, the very title seems to be problematic: “From Paul and Augustine to Mencius”. This does not give an expected chronological perspective! The Apostle Paul (c. 5 – c. 64/65 AD), Augustine of Hippo (13 November 354 – 28 August 430 AD), and Mengzi’s life dates are 372–289 BC! Thus, I suggest the title:
The Dilemma of Conscience: From Paul and Augustine with an Additional Excursus on Mencius
2. The topic of conscience as “the most hidden center in human being” is very important, also in the present-day debates on moral life. “Conscience” is a concept that has significant popularity even in liberal democratic societies. However, contemporary moral philosophy on conscience is surprisingly sparse. An exception is doing research on it in historical perspective, such as the study, e.g., St. Augustine.
3. From the “Introduction” should be seen a right perspective, i.e., the first two Christian thinkers – St. Paul and St. Augustine belong to Christianity – a religion that claims to have received a revelation from God (usually also recorded in writing). These revelation religions are primarily the monotheistic religions: Judaism, Christianity, and Islam with their respective holy scriptures: Hebrew Bible, Christian Bible, and Koran.
The Apostle Paul, who wrote a substantial part of the New Testament, wrote about the origin of his preaching, saying that he had received it as a revelation:
“11For I want you to know, brothers and sisters, that the gospel that was proclaimed by me is not of human origin; 12for I did not receive it from a human source, nor was I taught it, but I received it through a revelation of Jesus Christ.” Galatians 1:11-12
And what is the teaching of Mengzi?
As the second sage of Confucianism – after Confucius – is Mengzi a teacher of the pursuit of – especially – ethical knowledge and self-improvement of individuals, including the rulers. His theory of human nature purports that all human beings share an innate goodness that either can be cultivated through education and self-discipline or squandered through neglect and negative influences, but never lost altogether.
Thus, we have contrast between Christian soteriology and Confucian self-cultivation! Between God’s grace and human education! Already in the “Abstract,” the author says at the end in view of Mencius “… and the salvation of the fallen world.” In Confucianism there is no need for salvation, but only constant and tireless education and self-cultivation. Such a formulation “… and the salvation of the fallen world” is incorrect! The concept of Tian (Heaven) in the Mengzi should be deepened in the paper because this is a religious aspect of Confucianism and a pendent to the Christian God! All in all, Confucianism has no history of revelation, no prayers and tolerates other gods. Is it a religion?
4. “3. Augustin: Will, Grace and the Invention of Introspective Conscience” (pp. 4-6) – the problem with the word “invention” (first met in the “Abstract” and then some times throughout the text) is whether it is an invention and a discovery of introspective conscience.
Please consult the following article: “The Invention of Consciousness” by Nicholas Humphrey which supports the choice of the author!
Abstract: “In English we use the word “invention” in two ways. First, to mean a new device or process developed by experimentation, and designed to fulfill a practical goal. Second, to mean a mental fabrication, especially a falsehood, designed to please or persuade. In this paper I argue that human consciousness is an invention in both respects.
First, it is a cognitive faculty, evolved by natural selection, designed to help us make sense of ourselves and our surroundings. But then, second, it is a fantasy, conjured up by the brain, designed to change the value we place on our existence.”
https://link.springer.com/article/10.1007/s11245-017-9498-0 (download pdf.)
I would personally use the word “development” or “emergence.”
5. The Christian doctrine of justification (soteriological dimension) asks what must happen so that the relationship between man and God, which has been burdened by man’s sins, can be put right again.
The starting point of Paul’s doctrine of justification is that man inherits his sinfulness from Adam. Redemption only takes place through the atoning death of Christ. Christ gave his life to redeem the world. According to Paul, good works and all 613 regulations of the Jewish law are not enough to renew the inner man. Christ's substitutionary death ended the reign of the law, which was basically the purest righteousness of works. For Paul, the gospel comes first, at the center of which is faith in Jesus Christ and that the love of God frees man from sin.
The Augustinian doctrine of grace similarly assumes that all people are burdened with original sin (“peccatum haereditarium”) through Adam. Consequently, no human being is free from sin, so that every individual is doomed to death. The basic idea in this doctrine of grace is that all people inherit Adam’s sinfulness, which means eternal damnation for everyone. It is already clear that man cannot free himself from his sinfulness by his own efforts. According to the church father Augustine, only baptism frees people from original sin. This means that God, in his grace and through his mercy, redeems man. Redemption is not for everyone, but only for the elect. On the whole, it is clear that the Augustinian doctrine of grace contains rudiments of the doctrine of predestination.
“Mencius both stated his belief that he had recognised the Way of Heaven and was practicing it, and also announced his intention to save the corrupt and war-torn world through his Confucian wisdom and writings.” In the context of Christian thinking, Mencius would be a follower of … Pelagius (ca. 354–418: a Romano-British theologian). The basis of Pelagianism, which is to orthodox Christianity a heresy, is that original sin does not corrupt human nature (Mencius could have had no knowledge of this Christian controversy – so his view tends to a kind of natural Tian theology; that is important to provide Mencius’ understanding of Tian in his Mengzi), but that man has the ability to redeem himself (self-redemption). According to Pelagius, it can be concluded that man has a moral will, which enables him to distinguish between good and evil. Accordingly, Adam is a bad example of a human being and Jesus Christ is a good example of a human being. Furthermore, Pelagianism states that all of humanity bears full responsibility for its own salvation and sin. All in all, it can be concluded that man can bring about his own salvation through his own efforts by being without sin (“posse sine peccato esse,” meaning “to be able to be without sin”).
6. The critique of Mencius by Zhang Hao at the end of this article (p. 13) is abrupt and startling: “In his view, the expansion of conscience from inner sagehood to outer kingship argued by Mencius was never realized in ancient China and it neither initiated a democratic tradition in modern China; in contrast, this optimistic Confucian argument often became a vassal or accomplice of the autocratic monarchy (Zhang 2018, pp. 43-58; Shen 2011, pp. 133-144).” Yes, this moral epistemological optimism is a part of Confucianism. However, how it relates to the emergence or non-emergence of democracy is a different story.
It is a controversial issue. There some other views on this matter: Lary Lai, “Mencius and the New Confucianism’s Pursuit of Democracy”:
Abstract: “This chapter analyzes how the main ideas of Mencius’s political thought were interpreted, reconstructed, and appropriated by the three members of New Confucianism “xin rujia 新儒家” in the twentieth-century China, namely Xu Fuguan 徐復觀 (1904–1982), Tang Junyi 唐君毅 (1909–1978), and Mou Zongsan 牟宗三 (1909–1995). It presents the core textual analysis by the three thinkers in their interpretation of Mencius’s ideas, discusses these interpretations, and argues that Mencius was portrayed as pro-democracy and anti-monarchy in their writings. The chapter then concludes that the arguments that support the former image are doomed, and those that support the latter image are more convincing. Next, the chapter considers the set of Mencius-inspired arguments the thinkers hold in favour of democratic political participation. The thinkers attempted to conclude that democracy, which entails political subjectivity, is a more viable means to achieve the Confucian ends than a traditional monarchical system. The last part of this chapter argues that the thinkers’ arguments justify only a political account of individual subjectivity for democratic participation, which cannot explain why individuals as moral agents should take part in politics. A Mencian line of argument for an ethical account of political participation is therefore developed on the thinkers’ behalf.”
https://link.springer.com/chapter/10.1007/978-3-031-27620-0_15
7. Two corrections: 1) p. 2: bulü erzhi, better to write: bu lü er zhi
2) Chen Hao’s should be Cheng Hao程顥, 1032–1085, p. 10.
Suggestion: p. 11: Gaozi add Chinese characters and life dates 告子 (ca. 420-350 BC); the same above to Cheng Hao!
